# COVID-19 Elderly Patients Treated for Proximal Femoral Fractures during the Second Wave of Pandemic in Italy and Iran: A Comparison between Two Countries

**DOI:** 10.3390/medicina58060781

**Published:** 2022-06-09

**Authors:** Riccardo Giorgino, Erfan Soroush, Sajjad Soroush, Sara Malakouti, Haniyeh Salari, Valeria Vismara, Filippo Migliorini, Riccardo Accetta, Laura Mangiavini

**Affiliations:** 1Residency Program in Orthopedics and Traumatology, University of Milan, 20122 Milan, Italy; valeria.vismara@unimi.it; 2Faculty of Medicine and Surgery, University of Milan, 20122 Milan, Italy; erfan.soroush@unimi.it (E.S.); sajjad.soroush@unimi.it (S.S.); sara.malakouti@unimi.it (S.M.); haniyeh.salari@unimi.it (H.S.); 3Department of Orthopaedics, University Clinic Aachen, RWTH Aachen University Clinic, 52074 Aachen, Germany; migliorini.md@gmail.com; 4IRCCS Istituto Ortopedico Galeazzi, 20161 Milan, Italy; riccacc@gmail.com (R.A.); laura.mangiavini@unimi.it (L.M.); 5Department of Biomedical Sciences for Health, University of Milan, 20161 Milan, Italy

**Keywords:** SARS-CoV-2, proximal femoral fractures, traumatology, clinical features, second wave, Italy, Iran

## Abstract

*Background and objevtive*: The worldwide spread of SARS-CoV-2 has affected the various regions of the world differently. Italy and Iran have experienced a different adaptation to coexistence with the pandemic. Above all, fractures of the femur represent a large part of the necessary care for elderly patients. The aim of this study was to compare the treatment in Italy and Iran of COVID-19-positive patients suffering from proximal femur fractures in terms of characteristics, comorbidities, outcomes and complications. *Materials and Methods*: Medical records of COVID-19-positive patients with proximal femoral fractures treated at IRCCS Istituto Ortopedico Galeazzi in Milan (Italy) and at Salamat Farda and Parsa hospitals in the province of Tehran (Iran), in the time frame from 1 October 2020 to 16 January 2021, were analyzed and compared. *Results*: Records from 37 Italian patients and 33 Iranian patients were analyzed. The Italian group (mean age: 83.89 ± 1.60 years) was statistically older than the Iranian group (mean age: 75.18 ± 1.62 years) (*p* value = 0.0003). The mean number of transfusions for each patient in Italy was higher than the Iranian mean number (*p* value = 0.0062). The length of hospital stay in Italy was longer than in Iran (*p* value < 0.0001). Furthermore, laboratory values were different in the post-operative value of WBC and admission and post-operative values of CRP. *Conclusions*: The present study shows that differences were found between COVID-19-positive patients with proximal femoral fractures in these two countries. Further studies are required to validate these results and to better explain the reasons behind these differences.

## 1. Introduction

The worldwide spread of severe acute respiratory syndrome coronavirus 2 (SARS-CoV-2) has affected the various regions of the world differently, with diverse modes of diffusion and saturation of the health system [1]. During the first SARS-Cov-2 wave, Italy was one of the most hit countries; meanwhile, even in Iran, the spread of COVID-19 led to a dramatic situation from the perspective of management of the medical assistance activity [2,3,4]. These two nations have experienced a different adaptation to coexistence with the pandemic, profoundly changing the daily life activities and medical practice [5,6,7,8] by setting limitations and restrictions and adapting assistance activities to cope with the emergency [9,10,11]. The second wave, on the other hand, found both countries prepared to face the health emergency, having already allowed the opportunity to set up assistance protocols and organize a hierarchical priority in medical-assistance activities [12,13]. In particular, with the regular reopening of social and work activities, some pathologies have returned to play an important role in the request for management by health facilities. Furthermore, the new welfare organizations had to consider as ordinary those patients affected by COVID-19, which in itself guarantees a notable picture of comorbidity and care difficulties. It is well known that fractures of the femur represent a large part of the necessary care for elderly patients [14,15] and have been a fundamental item in the organization of the new protocols [16]. In the evaluation of these fragile patients, different research groups tried to better understand how COVID-19 infection impacted the management, morbidity and mortality of such patients [17,18,19,20]. Nevertheless, a direct comparison between different countries is lacking. With these assumptions, it was of particular interest to compare how these elderly and complex patients were treated in two countries that have faced the second wave of the pandemic. The aim of this study was to compare patients’ characteristics and treatment in Italy and Iran of COVID-19-positive elderly patients suffering from proximal femur fractures in terms of characteristics, comorbidities, outcomes and complications.

## 2. Materials and Methods

In this retrospective study, the medical records of COVID-19-positive patients with proximal femoral fractures treated at IRCCS Istituto Ortopedico Galeazzi in Milan (Italy) during the second wave of the pandemic, in the time frame from 1 October 2020 to 16 January 2021, were analyzed and compared with COVID-19-positive patients with proximal femoral fractures in the same timeframe treated at Salamat Farda and Parsa hospitals in the province of Tehran (Iran). Only COVID-19-positive patients were included in the study. Infection with SARS-COV-2 was detected in the majority of cases upon hospital arrival due to screening implemented by both the Italian and Iranian institutes with an RT-PCR test after nasopharyngeal swab. All patients underwent antithrombotic prophylaxis and were surgically treated. Data of each patient were collected from the medical records. Patients’ characteristics (such as age, sex, comorbidities, diagnosis, laterality), treatment, length of hospital stay, complications, oxygen support, transfusions, discharge mode and laboratory values (hemoglobin, hematocrit, platelet count, C-reactive protein, white blood cells and creatinine) were evaluated. Laboratory values were collected in three stages: upon admission, at 3 to 5 days after surgery and at discharge. Complications were collected and grouped according to the physio-pathological sphere into the following: cardiovascular, metabolic, respiratory, oncological, nephrological, neuropsychiatric and other (gastrointestinal, immunological).

The analysis was performed using SPSS software version 26 (IBM SPSS Statistics, Chicago, IL, USA). Categorical variables (number of patients in each group, surgical treatment, oxygen support) in the two groups were described using counts and percentages, whereas mean and standard error were used to report continuous variables (age, number of transfusions, length of hospital stay). Binomial tests were used to compare the two groups according to classification of comorbidities. Laboratory values (hemoglobin, hematocrit, platelet count, CRP, WBC and creatinine) at three different time intervals were compared among the two groups using the unpaired Student’s t-test to evaluate the normal data distribution. Significance was set at *p* value < 0.05.

## 3. Results

The COVID-19 Italian group was composed of 37 patients (10 males and 27 females), while the Iranian group was composed of 33 patients (10 males and 23 females). The Italian group was statistically older than the Iranian group, respectively, with a mean age of 83.89 ± 1.60 years vs. 75.18 ± 1.62 years (*p* value = 0.0003). All proximal femoral fractures were surgically treated. More precisely, in Italy, 1 total hip arthroplasty (THA), 10 bipolar hemiarthroplasties (BH) and 26 proximal femoral nails (PFN) were performed. In Iran, 5 patients were treated with THA, 15 with BH, 6 with PFN and 7 with dynamic hip screws (DHS). In both groups, cardiovascular comorbidities were the most frequent. In Italy, 29 patients had cardiovascular comorbidities, 14 metabolic, 4 respiratory, 6 oncological, 2 nephrological, 8 neuropsychiatric, 2 other comorbidities, and 1 patient had none. In Iran, 17 patients had cardiovascular comorbidities, 15 metabolic, 4 respiratory, 1 oncological, 6 nephrological, 3 neuropsychiatric, 4 other comorbidities, and 2 patients had none. Upon admission, while only 18 patients in Italy (48.6%) were treated with oxygen support, all Iranian patients were treated with oxygen support (100%). The mean number of transfusions for each patient in Italy was 3.08 ± 0.50, and it emerged as statistically higher than the Iranian mean number of 2.12 ± 0.92 (*p* value = 0.0062). The length of hospital stay in Italy was longer than in Iran, with 13.24 ± 1.18 days vs. 4.27 ± 0.32 days (*p* value < 0.0001). All patients survived surgery in the early post-operative period. When focusing our attention on laboratory values (hemoglobin, hematocrit, platelet count, C-reactive protein, white blood cells and creatinine), comparisons of the two groups upon admission, 3/5 post-operative day and discharge are presented in Table 1.

The post-operative value of white blood cells in the Italian group was statistically higher than in the Iranian sample, with a mean value of 11.50 ± 1.46 vs. 9.79 ± 0.62 (*p* value = 0.0288) (Figure 1).

The admission value of CRP in the Italian group was statistically higher than in the Iranian one, with a mean value of 5.40 ± 0.87 vs. 2.50 ± 0.19 (*p* value = 0.0025). The same statistical difference is still present at the post-operative values (7.89 ± 1.28 vs. 3.41 ± 0.24 (*p* value = 0.0014), as can be seen in Figure 2.

## 4. Discussion

The main clinical relevance of the present study resides in the analysis of the differences in the management of important pathologies in the assistance activity of orthopedics, such as proximal femur fractures, between two different countries during a pandemic never seen before. According to the main findings of this study, many differences were found through the analysis of these two groups. The Italian group was statistically older than the Iranian group, with a longer length of hospital stay and a higher mean number of transfusions for each patient in Italy. Furthermore, concerning laboratory indices, WBC post-operative values and both CRP admission and post-operative values were significantly different. As mentioned above, the length of hospital stay in Italy was longer than in Iran, with 13.24 ± 1.18 days vs. 4.27 ± 0.32 days (*p* value < 0.0001). In particular, it should be emphasized that the patients of the Iranian group were hospitalized in private facilities, and therefore, the length of the hospital stay was undoubtedly influenced by the sudden transfer to physiotherapy wards for several reasons, including economic ones. Moreover, most rehabilitative structures in Italy, to avoid the spread of the SARS-COV-2 infection, avoid taking patients in until a negative test is available, thereby increasing the hospital length of stay. Nevertheless, encouraging evidence in the literature suggests that an enhanced recovery after surgery (ERAS) protocol can benefit patients with hip fractures [21], but undoubtedly, in the case of a patient suffering from COVID-19, the matter becomes more complicated by having to interface with further timelines linked to specific anti-COVID protocols. This is an important aspect, as we have already previously described how our hospital was able to improve some parameters of hospital clinical efficiency during the first wave of the pandemic. Indeed, in the study of Brayda-Bruno et al., the time frames from diagnosis to surgery and from diagnosis to discharge were analyzed, reporting a reduction in diagnosis–discharge time [22]. The increased length of stay of the Italian patients could even be related to the higher age in the Italian group, as already underlined in the literature, where older age is correlated with a longer length of stay [23,24]. However, other studies did not report a correlation between advanced age in patients with femoral fractures and a prolonged hospital stay [25,26].

Regarding the laboratory data shown in the table, there are some considerations to emphasize. In our results, we found that post-operative value of WBC was statistically higher in the Italian group compared to the Iranian patients. It has been reported in the literature that among trauma patients, higher WBC values were detected in patients with major lesions rather than in patients with minor lesions, but both were within the normal range [27]. These data should be integrated with what emerges from recent studies on COVID-19 patients, where it was described that WBC count in COVID-19 patients is normal or slightly reduced in the early stages and that these values may change as the disease progresses [28]. Furthermore, recent studies suggest that various parameters, including WBC, could predict critical disease progression [29]. Another result that emerged from our study was the finding of higher values of CRP in the Italian group upon admission and post-operative time. For all COVID-19 patients, we should consider a high level of inflammation due to the nature of COVID-19 disease [30,31,32]. Indeed, in the study of Puzzitiello et al., it is reported how these patients with orthopedic trauma injuries may have an amplified response to the traumatic insult because of their baseline hyperinflammatory and hypercoagulable states [33]. The same findings were confirmed by Bayrak et al., where inflammatory parameters, including CRP, were higher in COVID-19-positive patients with femoral fractures [34]. Our results are adequately framed within the literature findings; however, further studies should be conducted to explain the reason for the laboratory differences between the patients of the two countries.

This study presents several limitations. First of all, it is a case series. Second, the collection of clinical documentation was conducted in two different countries in line with the primary objective of this work in the study of differences. However, this collection of data had to face different pre-operative, surgical and post-operative treatment protocols, as already illustrated above. In this respect, it is not easy to compare the pre-operative preparation and surgical technique of different surgeons from two different countries. Third, there is still little literature about the differences of COVID-19-positive patients among different countries, and future studies should investigate any differences and points in common in order to establish the best possible treatment throughout the duration of the medical intervention. Furthermore, possibly, further studies should increase the sample size, which, in the present work, is already important, considering the sampling of patients. Another limitation is scarce information about the COVID-19 infection of these patients (onset of infection, severity, quantity of oxygen support needed, relationship with patient outcomes). Due to the above-mentioned difficulties in collecting data, some records about the onset of the viral infection and the symptoms are lacking. In particular, we do not know if the traumatic event might have been directly related to the COVID-19 infection (i.e., due to muscular weakness); it is also highly possible that our patients were hospitalized and treated in different stages of the viral infection, thus affecting the evaluated outcomes. Finally, another important aspect that constitutes a limitation resides in the follow-up of these patients, which is limited to the few days of hospital stay in the post-operative period. Further studies should be able to evaluate the survival of COVID-19 patients surgically treated precisely in order to perhaps be able to draw up international guidelines.

## 5. Conclusions

The present study shows that some differences were found between COVID-19-positive patients with proximal femoral fractures in Italy and Iran in terms of mean age, length of hospital stay, number of transfusions, WBC count and CRP values. Further studies are required to validate these results, to better explain the reasons behind these differences and to establish the best treatment throughout the duration of the medical intervention.

## Figures and Tables

**Figure 1 medicina-58-00781-f001:**
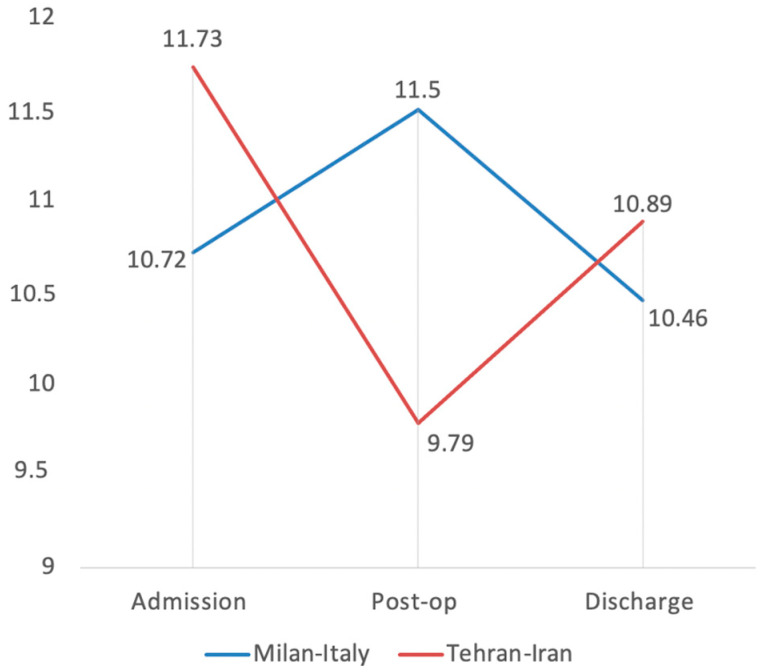
Trend of white blood cells (WBC) in COVID-19 Italian and Iranian groups.

**Figure 2 medicina-58-00781-f002:**
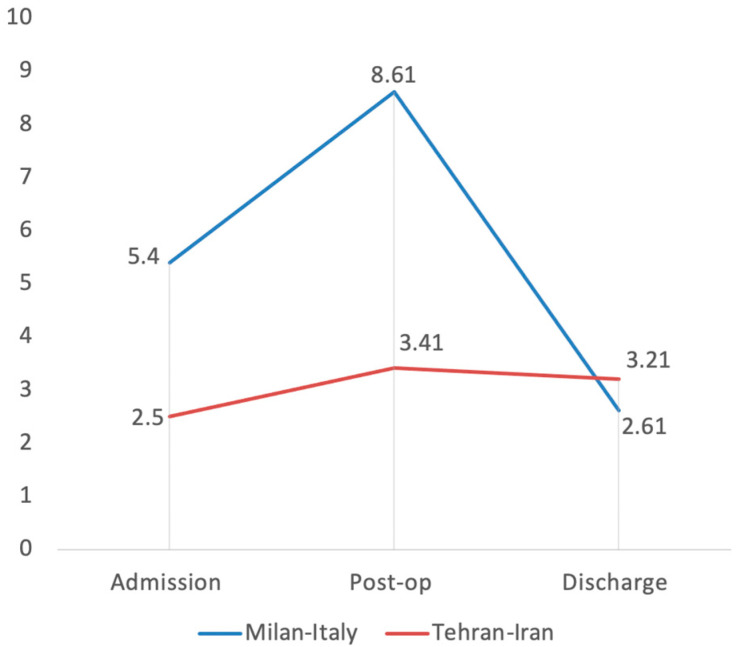
Trend of C-reactive protein (CRP) in COVID-19 Italian and Iranian groups.

**Table 1 medicina-58-00781-t001:** Comparison between laboratory values of COVID-19 Italian and Iranian groups upon admission, 3–5 post-operative day and discharge.

Lab Values	Admission		3–5 Postoperative Day		Discharge	
	Italy	Iran	Italy	Iran	Italy	Iran
Hemoglobin (g/dL)	11.90 ± 035	12.24 ± 0.31	10.24 ± 0.22	10.61 ± 0.25	11.31 ± 0.71	10.01 ± 0.27
Hematocrit (%)	36.99 ± 1.02	35.88 ± 1.02	31.79 ± 0.68	31.30 ± 0.76	33.75 ± 0.81	30.31 ± 0.88
Platelet count (103/μL)	265.41 ± 19.48	269.45 ± 18.64	276.84 ± 16.44	249.57 ± 16.56	321.12 ± 31.84	285.07 ± 18.94
CRP (mg/L)	5.40 ± 0.87	2.50 ± 0.19	7.89 ± 1.28	3.41 ± 0.24	4.79 ± 0.97	3.21 ± 0.18
WBC (103/μL)	10.72 ± 1.11	11.73 ± 1.09	11.50 ± 1.46	9.79 ± 0.62	10.46 ± 0.93	10.89 ± 0.59
Creatinine (mg/dL)	0.96 ± 0.06	1.02 ± 0.07	1.22 ± 0.30	1.04 ± 0.09	0.83 ± 0.07	0.96 ± 0.09

## Data Availability

Data supporting the reported results can be found in database generated during the study.

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
