# Peer review of "COVID-19 Elderly Patients Treated for Proximal Femoral Fractures during the Second Wave of Pandemic in Italy and Iran: A Comparison between Two Countries"

_medicina, 2022, doi:10.3390/medicina58060781_

Round 1
Reviewer 1 Report
This study aimed to compare the treatment in Italy and Iran of COVID-19-positive patients suffering from proximal femur fractures in terms of characteristics, comorbidities, outcomes, and complications in a small sample. The paper is well written, albeit its clinical significance remains unclear.
Major concerns
1. What is the main clinical relevance of the results?
Minor concerns
1. Was the severity of COVID infection taken into account in the analysis? (CO-RADS score, CT severity score)? The severity and the bacterial superinfection of the COVID can bias the findings (CRP, WBC).
2. Was the same treatment used for the COVID infection with the same drugs? If not, how could it influence the results? 3. Was the osteoporosis known in the patients? How was the Vitamin D level of the patients which could influence both COVID and osteoporotic fracture? 4. Was the lab result influenced by postoperative infections? How was the infection control ensured in both countries?Author Response
Dear Reviewer, thanks a lot for your wise comments.
Major concerns
- The main clinical relevance of these results resides in the analysis of the differences in the management of important pathologies in the assistance activity of orthopedics such as proximal femur fractures between two different countries, during a pandemic never seen before. Our study shows significant differences between the two groups in terms of mean age, length of hospital stay, number of transfusions, WBC count and CRP values. In our opinion the length of hospital stay represents the main result, underlying a different approach in the management of COVID-19 positive patients with proximal femoral fractures due to several reasons (line 135-152). We really appreciate your comment and decided to emphasize this concept at the beginning of the discussion, modifying the paper as required.
Minor concerns
- We have reported an indirect sign of severity of COVID-19 infection ad admission. In Italy only 18 under 37 patients (48,6%) were treated with oxygen support, while all Iranian patients were treated with oxygen support (100%) (line 104-105). We have radiological examinations of the Italian group but no informations about the Iranian one, so we decided to report only the need of oxygen support. Even in this case, we really appreciate your comment and decided to emphasize this concept adding the percentage of oxygen support.
- Even if all patients underwent antithrombotic prophylaxis, undoubtedly the difference in pharmacological treatment represents one of the main limitations of this study, as highlighted in the section of limitations (line 177-180).
- Unfortunately, osteoporosis and Vitamin D levels were not known in the patients. These informations could influence both COVID and osteoporotic fracture and further studiers are required to go into detail it.
- Unfortunately, we are in possession of informations only on how the infection control was ensured in Italy. As mentioned in the discussion, in the study of Brayda-Bruno et al. we have described how our hospital was able to improve some parameters of hospital clinical efficiency through specific anti-covid protocols (line 148-150).

Reviewer 2 Report
Interesting and relevant study.
However, I am curious why compare to Iran and not other country? Any of the collaborators were Iranian? Please declare this.
Author Response
Dear Reviewer, thanks a lot for your work and your comment.
The comparison was made with Iran because two collaborators were from Iran and provided their data.